# Stearoyl-CoA Desaturase Is Essential for Porcine Adipocyte Differentiation

**DOI:** 10.3390/ijms21072446

**Published:** 2020-04-01

**Authors:** Lulu Liu, Yu Wang, Xiaojuan Liang, Xiao Wu, Jiali Liu, Shulin Yang, Cong Tao, Jin Zhang, Jianhui Tian, Jianguo Zhao, Yanfang Wang

**Affiliations:** 1Laboratory of Animal (Poultry) Genetics Breeding and Reproduction, Ministry of Agriculture, Institute of Animal Science, Chinese Academy of Agricultural Sciences, Beijing 100193, China; 2College of Animal Science and Technology, China Agricultural University, Beijing 100193, China; 3Institute of Zoology, Chinese Academy of Sciences, Beijing 100101, China; 4Biotechnology Research Institute, Shanghai Academy of Agricultural Sciences, Shanghai 201106, China; 5College of Biological, Chemical Sciences and Engineering, Jiaxing University, Jiaxing 314001, China; 6Savaid Medical School, University of Chinese Academy of Sciences, Beijing 100049, China; 7College of Life Sciences, Qingdao Agricultural University, Qingdao 266109, China

**Keywords:** *SCD*, adipocytes, transdifferentiation, adipogenesis

## Abstract

Fat deposition, which influences pork production, meat quality and growth efficiency, is an economically important trait in pigs. Numerous studies have demonstrated that stearoyl-CoA desaturase (SCD), a key enzyme that catalyzes the conversion of saturated fatty acids into monounsaturated fatty acids, is associated with fatty acid composition in pigs. As *SCD* was observed to be significantly induced in 3T3-L1 preadipocytes differentiation, we hypothesized that it plays a role in porcine adipocyte differentiation and fat deposition. In this study, we revealed that *SCD* is highly expressed in adipose tissues from seven-day-old piglets, compared to its expression in tissues from four-month-old adult pigs. Moreover, we found that *SCD* and lipogenesis-related genes were induced significantly in differentiated porcine adipocytes. Using CRISPR/Cas9 technology, we generated *SCD^-/-^* porcine embryonic fibroblasts (PEFs) and found that the loss of *SCD* led to dramatically decreased transdifferentiation efficiency, as evidenced by the decreased expression of known lipid synthesis-related genes, lower levels of oil red O staining and significantly lower levels of triglyceride content. Our study demonstrates the critical role of *SCD* expression in porcine adipocyte differentiation and paves the way for identifying it as the promising candidate gene for less fat deposition in pigs.

## 1. Introduction

Fat deposition is the key economic trait in pig production that affects growth rate, pork production and meat quality [1,2]. Genetic improvement towards less fat deposition and high meat quality has been the focus of pig breeding programs for a long time. With the rapid development of high-throughput screening technologies, numerous candidate genes and noncoding RNAs that may affect the traits of fat deposition and fatty acid (FA) composition in pigs have been screened over recent decades [3,4,5]. However, the exact biological function of these candidate genes in fat metabolism needs to be further investigated.

Stearoyl-CoA desaturase (SCD), a well recognized rate-limiting enzyme in the biogenesis of endogenous monounsaturated fatty acids (MUFAs), mainly oleate and palmitoleate, has been reported to serve a protective role to allow for continued MUFAs synthesis during dietary unsaturated fat insufficiency [6]. Accumulating evidence from human and animal models has revealed the important roles of SCD in the regulation of systemic lipid metabolism, especially in metabolic tissues [7]. Notably, global SCD1 knockout (KO) mice were protected against diet-induced adiposity, showing increased insulin sensitivity and energy expenditure [8]. Liver-specific SCD1*-*KO mice were protected from high carbohydrate-induced obesity and hepatic steatosis, because of reduced de novo lipogenesis and decreased fatty acid synthesis [9]. The results from these studies demonstrate the critical roles of SCD1 in regulating fat deposition and lipid metabolism. Furthermore, the molecular regulatory mechanisms of *SCD1* have also been explored. It has been reported that *SCD* is transcriptionally regulated by the binding of sterol regulatory element-binding protein-1c (SREBP-1c) to the specific sterol regulatory element (SRE) of *SCD* [10].

*SCD* was found to be associated with the FA composition and backfat thickness candidate genes in pigs by genome-wide association studies (GWAS) [11] and quantitative trait locus (QTL) analysis [12]. The single nucleotide polymorphisms (SNPs) in porcine *SCD* genes, especially two mutations in the promoter region, g.-353 C > T and g.-233 T > C, were also identified [12], and their relationship with fatty acid composition has been investigated in the Duroc purebred population and the Korean native pig × Landrace F_2_ crossbreed [13]. The association of porcine *SCD* SNPs with intramuscular fat content was also found in Laiwu pigs and commercial pig breeds [14]. Furthermore, the expression of *SCD* alleles showed significant allelic imbalance in adipose tissue, and *SCD* was a potential molecular target for regulating porcine fatness traits [15]. Based on these observations, we hypothesized that SCD plays a key role in porcine adipocyte differentiation.

In this study, we examined the expression levels of SCD in adipose tissues from seven-day-old piglets and four-month-old adult pigs of the Bama native Chinese pig breed that have a high capacity for fat deposition. We also examined SCD expression during adipocyte differentiation. Furthermore, we generated *SCD*-KO porcine embryonic fibroblasts (PEFs) using the CRISPR/Cas9 method and investigated their role in adipogenesis.

## 2. Results

### 2.1. SCD Exhibited a Higher Expression Level in Adipose Tissues from Bama Piglets

It has been reported that adipose tissues from piglets contain more adipose tissue-derived mesenchymal stem cells [16]. To examine the expression levels of *SCD* in adipose tissues in piglets and adult pigs, we collected subcutaneous adipose tissue from both the dorsal and leg, perirenal adipose tissue and leaf adipose tissue from Bama pigs at seven days and four months of age and performed a real-time PCR analysis. Our data showed that *SCD* had higher expression levels in each adipose depot from seven-day-old Bama piglets, compared to those from four-month-old Bama adult pigs, but only the differences in leg subcutaneous adipose tissue reached statistical significance (Figure 1A). This observation was confirmed by Western blot analysis results (Figure 1E). We are also interested in detecting the expression level of transcriptional factors that regulate lipogenesis, including *PPARG*, *C/EBPA* and *SREBP-1C*. Our real-time PCR data showed that *PPARG* had a significantly higher expression in adipose tissues from seven-day-old Bamapiglets, relative to those from adult pigs (Figure 1B). In addition, dorsal subcutaneous adipose tissue and leaf adipose tissue from seven-day-old piglets exhibited a significantly higher level of *C/EBPA* and *SREBP-1C*, while no significant differences of gene expression were observed in the other two depots between two groups of pigs (Figure 1C,D).

### 2.2. SCD Expression was Significantly Up-Regulated in Differentiated Porcine Adipocytes

To examine whether *SCD* is involved in porcine adipocyte differentiation, we primarily cultured adipose tissue-derived mesenchymal stem cells from piglets at ten days of age. *In vitro* differentiation was performed, and obvious lipid droplets were observed by oil red O staining (Figure 2A). The expression of SCD, along with the well known adipocyte differentiation markers PPARG and C/EBPA [17,18], was elevated in mature adipocytes (Figure 2B). In addition, the real-time PCR results revealed the significant elevation of lipogenesis related genes *FASN* and *ELOVL6*, in the mature adipocytes, while the expression level of *FADS2* was not changed (Figure 2C). To examine the dynamic changes in expression levels of SCD during differentiation, cell lysates were collected at different time points and subjected to Western blot. Our data showed that SCD was induced at the late stage of differentiation (Appendix A). Collectively, our data showed that *SCD* was highly induced in differentiated porcine adipocytes.

### 2.3. Generation of SCD^-/-^ Porcine Embryo Fibroblasts

To investigate the effect of *SCD* on lipogenesis, we generated *SCD-*KO porcine embryo fibroblasts (PEFs) by the CRISPR/Cas9 technique. First, four different sgRNAs (sgRNA1-4) that target exon 2 in the porcine *SCD* gene were designed by an online tool (http://crisps.mit.edu/), based on the sequence of the porcine *SCD* gene (NCBI No. NC_010456.5). Figure 3A shows the gene structure of *SCD* and the sgRNAs, which are highlighted in blue. The sgRNAs were inserted into the Cas9/gRNA (puro-GFP) vector. Three pairs of sgRNAs, which were selected based on a potential shift mutation by deletion in the targeted fragment of each sgRNA, were co-transfected into the cells, and PCR was performed. Our data showed that the pair of sgRNA1- and sgRNA4-transfected cells produced a short band (Figure 3B), suggesting effective gene targeting, and this pair of sgRNAs was used for the following study.

The vectors containing sgRNA1 and sgRNA4 were co-transfected into PEFs, and after 48 h of transfection, the single cells were sorted by flow cytometry and seeded into 96-well plates. We selected 20 single-cell colonies randomly for PCR amplification (Figure 3C). The PCR products were subjected to sequencing, and our data showed that 5 colonies had biallelic mutations, with a targeting efficiency of 25% (Figure 3D). Figure 3E shows the sequencing data of the #9 colony, in which both alleles were modified by a 1 bp insertion and 140 bp deletion (Figure 3E). Furthermore, to detect the gene knockout efficiency, we examined the expression level of *SCD* in wild-type and #9 colony by real-time PCR. Our data showed that *SCD* was almost undetectable in the #9 colony (Figure 3F), suggesting successful *SCD* knockout. Hereafter, the #9 colony are referred to as *SCD^-/-^* cells, and the wild-type PEFs are named *SCD^+/+^* cells.

### 2.4. SCD Knockout Inhibited PEF Transdifferentiation into Mature Adipocytes

It was unclear whether PEFs can be transdifferentiated into mature adipocytes. Thus, we used the same protocol as described above, to differentiate the PEFs into mature adipocytes. The evidence from oil red O staining confirmed that the PEFs could be differentiated into mature adipocytes (Appendix A), and adipogenesis was further confirmed by the induction of the transcription factors PPARG and C/EBPA (Appendix A). Then, we differentiated *SCD^+/+^* and *SCD^-/-^* cells into mature adipocytes. Oil red O staining revealed the dramatically reduced differentiation efficiency of the *SCD^-/-^* cells (Figure 4A). The Western blot analysis results showed that *SCD* expression was undetectable in the differentiated *SCD*^-/-^ cells (Figure 4B). In addition, we examined the expression of lipogenesis-related genes, namely, *SREBP-1C*, *PPARG*, *C/EBPA*, *FASN*, *ELOVL6*, *FADS2*, *DGAT1*, *DGAT2*, *FABP4* and *ACACA*, in both the *SCD^+/+^* and *SCD^-/-^* cells by real-time PCR. Our data showed that their expression was significantly lower in the *SCD*^-/-^ cells than it was in *SCD*^+/+^ cells (Figure 4C). We also measured the triglyceride (TG) content and non-esterified fatty acid (NEFA) content in the medium of the *SCD^+/+^* and *SCD^-/-^* cell cultures, and found that both TG and NEFA were significantly decreased in the *SCD*^-/-^ cells (Figure 4D,E). Furthermore, we also measured the expression levels of the genes involved in fatty acid *β*-oxidation, carnitine palmitoyltransferase 1a (CPT1A), carnitine palmitoyltransferase 2 (CPT2), and very long chain acyl-CoA dehydrogenase (VLCAD), and all three genes were up-regulated in the *SCD*^-/-^ cells, but the difference did not reach the significance level for the *VLCAD* gene (Figure 4F).

## 3. Discussion

Fat deposition is a very important economic trait that determines pig production, feed efficiency and meat quality, including flavor and tenderness [1,2,19]. As a well-known candidate gene involved in the regulation of intramuscular FA content and meat quality, SCD was shown by our data to also be a key enzyme for the control of adipogenesis. Consistent with previous observations in 3T3-L1 preadipocytes [20] and bovine preadipocytes [21], SCD was found to be induced to high levels during porcine adipocyte differentiation. Moreover, our data demonstrated that SCD deprivation has the potential to compromise the ability of PEFs to form lipids, suggesting that SCD is required for adipogenesis.

*SCD* expression has been previously recognized as a marker of adipocyte terminal differentiation [22], therefore, we expected that the adipose tissues from adult pigs would contain higher expression levels of *SCD*. However, we observed that the expression level of *SCD* was much higher in the adipose tissue from the seven-day-old Bama piglets than it was in the adipose tissue from the four-month-old Bama adult pigs. It has been reported that adipose tissues from piglets contain more adipose tissue-derived mesenchymal stem cells and the multipotency of pig adipose-derived stem cells decreases as the pig ages [16]. Based on the observation that *SCD1* was induced during the differentiation of 3T3-L1 pre-adipocytes [20] and porcine pre-adipocytes, we speculate that *SCD* was significantly induced in differentiating adipocytes and might be also considered as the adipocyte differentiation marker. In addition, Smith et al. designed experiments to examine adipose *SCD* expression levels in preweaning and postweaning obese and crossbred pigs. Despite the observation that *SCD* was negligibly detected on 0 days of age and increased significantly through 49 days of age, and more *SCD* mRNA was found in the adipose tissue from obese pigs than in the crossbred pigs during the suckling period. The *SCD* expression pattern was totally different in the adipose tissues of postweaning pigs, as the crossbred pigs exhibited greater *SCD* gene expression than did the obese pigs during the postweaning period [22]. Taken together, these data indicated that the regulation of SCD expression during different developmental stages and in various fat depots in pigs is complex, and SCD expression is very sensitive to diet; thus, more systemic, well designed experiments are needed to explore its expression and biological function in the adipose tissues of pigs.

*SCD* has been reported to be a target gene of the well known lipogenesis transcriptional factor sterol regulatory element-binding protein-1C (SREBP-1C) and is positively regulated by sterol regulatory binding elements in the UTR region of *SCD* [7]. Here, our data showed that SCD ablation significantly decreased the expression level of SREBP-1C, which was consistent with the observation that SCD1-deficient mice showed a decrease in the expression and maturation of SREBP-1 [23] and that *SCD*-deficient goat mammary epithelial cells had reduced triacylglycerol content and decreased *SREBP1*, *FASN*, *FABP3* and *FABP4* expression [24]. It has been shown that oleate positively regulates the expression and activity of SREBP-1c [23]. We speculate that, as a key enzyme in the biogenesis of oleate and palmitoleate [7], SCD ablation decreased the production of oleate and then down-regulated the expression of SREBP-1C, and further down-regulated other lipogenesis genes, namely, *PPARG*, *C/EBPA*, *FASN*, *ELOVL6, FADS2*, *DGAT1*, *DGAT2* and *FABP4*, and finally suppressing lipogenesis. Similar data were observed regarding the regulatory role of SCD on SREBP-1C expression via oleate, and they need to be validated experimentally.

It has been reported that the global disruption of SCD1 in mice reduces body adiposity and increases insulin sensitivity, making mice resistant to diet-induced weight gain [8]. Recently obtained data revealed the unappreciated role of SCD1 in potentiating beige adipocyte formation in mice [25]. These observations suggest that *SCD* is a promising candidate gene for reducing fat deposition in pigs, and the mechanisms that precisely regulate the SCD activity in adipose tissues need to be further explored.

## 4. Materials and Methods

### 4.1. Animals and Tissue Collection

Adult Bama pigs (four-month-old) and piglets (seven-day-old) (*n* = 3/group) were used in this study. Adipose tissues, including subcutaneous adipose tissues from dorsal and leg, perirenal aidpose tissues and leaf adipose tissues, were collected and snap frozen at −80 °C for later use. All animal experiments were performed according to the procedures approved by the Institutional Animal Care and Use Committee of the Institute of Animal Science, Chinese Agricultural Academy of Sciences (CAAS, Beijing, China).

### 4.2. Porcine Preadipocyte Pprimary Culture and In Vitro Differentiation

We collected adipose tissues from ten-day-old Bama pigs and washed them in Dulbecco’s phosphate-buffered saline (DPBS, Corning, Corning, NY, USA), with 5% penicillin-streptomycin. The adipose tissues were cut into pieces and digested in a cell incubator with 2 mg/mL collagenase I (Sigma, St. Louis, MO, USA). Then, the mixture was filtered with a 100 μm cell strainer (Falcon, New York, NY, USA) and centrifuged at 1500 rpm for 5 min. Porcine preadipocyte cells were plated in 100 mm cell culture dishes and grown in Dulbecco’s modified Eagle’s medium (DMEM/F12, LONZA, Basel, Kanton Basel, Switzerland), supplemented with 10% FBS and 1% penicillin-streptomycin.

Porcine preadipocyte cells were treated with the same method used for differentiation [26]. Two days after being cultured to confluence, the cells were induced with differentiation medium, consisting of DMEM supplemented with 10% FBS, 20 μM HEPES (pH 7.4, Thermo Fisher Scientific, Waltham, MA, USA), 33 μM biotin (Sigma, St. Louis, MO, USA), 17 μM pantothenic acid, 5 μg/mL insulin, 1 μM dexamethasone (Sigma, St. Louis, MO, USA), 0.25 mM isobutylmethylxanthine (Sigma, St. Louis, USA), and 1 μM rosiglitazone (Sigma, St. Louis, MO, USA), for 5 days. Then, one-half of the differentiation medium was removed, and one-half of the maintenance medium was added. The maintenance medium consisted of Dulbecco’s modified Eagle’s medium (DMEM, LONZA, Basel, Kanton Basel, Switzerland), supplemented with 10% FBS, 20 μM HEPES, 33 mM biotin, 17 mM pantothenic acid, 5 μg/mL insulin and 1 μM dexamethasone, and this was used to incubate the cells for 1 day. After 6 days in differentiation medium, the medium was replaced with maintenance medium, and the cells were incubated for another 2 days. PEFs were transdifferentiated into adipocytes according to the same method. Oil red O staining was used to stain the lipid droplets on differentiation day 8.

The PEFs were kept in JianguoZhao’s laboratory at the Institute of Zoology, Chinese Academy of Sciences (CAS, Beijing, China). These PEFs were maintained in DMEM, supplemented with 20% fetal bovine serum (FBS, HyClone, Logan, UT, USA) and 1% penicillin-streptomycin (Gibco, Grand Island, NY, USA). Porcine preadipocyte cells and PEFs were incubated at 38 °C in 5% CO_2_.

### 4.3. Gene Targeting by the CRISPR/Cas9 System

The Cas9/sgRNA (puro-GFP) vector was purchased from ViewSolid Biotech (No. VK001-02, ViewSolid Biotech, Beijing, China). Four sgRNAs that target exon 2 in porcine *SCD* were designed using an online tool (http://crispr.mit.edu/) (named sgRNA1, sgRNA2, sgRNA3 and sgRNA4) and cloned into the Cas9/sgRNA (puro-GFP) vector. Three pairs of sgRNAs, namely, sgRNA1 and sgRNA2, sgRNA1 and sgRNA4, and sgRNA2 and sgRNA3, were selected for co-transfecting into the PEFs by nucleofection using the Lonza basic fibroblasts nucleofector kit (LONZA, Basel, Kanton Basel, Switzerland), based on their potential for shift mutations. Then, after transfection for 48 hours with the appropriate sgRNA combination, the transfected PEFs were subjected to fluorescence activated cell sorting (FACS), based on green fluorescent protein (GFP) fluorescence. Each cell sorted by FACS was seeded into 96-well plates and cultured for 8 days. Cell colonies were cultured to confluence, and the efficiency of the sgRNA transfection was detected by PCR. The locations where the primers were expressed are shown in Figure 2A, and the expected PCR product is 403 bp. The best sgRNA combination was used for co-transfecting the PEFs, and the cells were sorted by FACS. PCR products from the cell colonies were sequenced to determine biallelic mutant colonies. To prevent the deletion of large fragments in cell colonies by using the CRISPR/Cas9 system, we designed another primer pair to identify the sequence of cell colonies. The locations of the primers are also shown in Figure 2A, and the expected PCR product is 1686 bp.

### 4.4. Real-Time PCR

Total RNA was extracted from tissues and cells using TRIzol reagent (Invitrogen, Waltham, MA, USA), and 1 μg of each RNA sample was reverse transcribed to cDNA using the First Strand cDNA synthesis kit (Thermo Scientific, Waltham, MA, USA). Real-time PCR was performed using a 7500 Fast Real-Time PCR system (Applied Biosystems, Foster, CA, USA), and the reaction was carried out in 20 μL containing SYBR Green (RR420A, TaKaRa, Tokyo, Japan). The relative expression of the genes was normalized to that of the control gene *GAPDH* and analyzed by the 2^–^^△△CT^ method. The primers are listed in Appendix A.

### 4.5. Western Blotting

Protein lysates from tissues and cells were harvested as previously described [27]. Equivalent amounts of the proteins were separated on 10% SDS-PAGE gels and transferred to nitrocellulose membranes. The membranes were blocked for 1 hour with 5% fat-free milk and incubated with primary antibodies overnight at 4 °C. The following primary antibodies were used: anti-SCD (ab39969, 1:2000, Abcam, Cambridge, USA), anti-TUBULIN (2146S, TUBULIN, 1:2000, CST, Danvers, MA, USA), anti-PPARG (81B8, 1:2000, CST, Danvers, MA, USA), and anti-C/EBPA (D56F10, 1:2000, CST, Danvers, MA, USA). The next day, the membranes were incubated with HRP-linked secondary antibodies (7074S, 1:2000, CST, Danvers, MA, USA) for 40 min. The signals were detected using SuperSignal^TM^ West Dura Extended Duration Substrate (Thermo Scientific Pierce, Waltham, MA, USA).

### 4.6. Oil Red O Staining

The cells were washed with DPBS 3 times and fixed in 4% paraformaldehyde solution for 15 min. Oil red O (Sigma, St. Louis, MO, USA) staining solution (60% stock solution and 40% deionized water), incubated at room temperature for 10 min before use. The cells were briefly washed with deionized water and rinsed once with 60% isopropanol. Oil red O staining solution was added to the plate and incubated for 10 min. The cells were rinsed with 60% isopropanol to redissolve the oil red O. A microscope (TH4-200, Olympus, Tokyo, Japan) was used to observe oil red O-stained cells.

### 4.7. TG and NEFA Measurements

The triacylglycerol (TG) content was measured with enzymatic assay kits from Applygen (Beijing, China). The non-esterified fatty acid (NEFA) content was measured using colorimetric assays (Wako Chemical, Osaka, Japan).

### 4.8. Statistical Analysis

The statistical analysis was performed with GraphPad Prism 6.0 (GraphPad software, La Jolla, CA, USA). All the data were from three biological replicates. Comparisons between groups were performed using two-tailed Student’s *t* test. Data are shown as the means ± SEM. *p*-Values of * *p* < 0.05, ** *p* < 0.01 and *** *p* < 0.001 were considered significant.

## 5. Conclusions

In conclusion, we found that porcine adipocyte differentiation requires SCD expression and highlighted the important contribution of SCD to porcine adipogenesis.

## Figures and Tables

**Figure 1 ijms-21-02446-f001:**
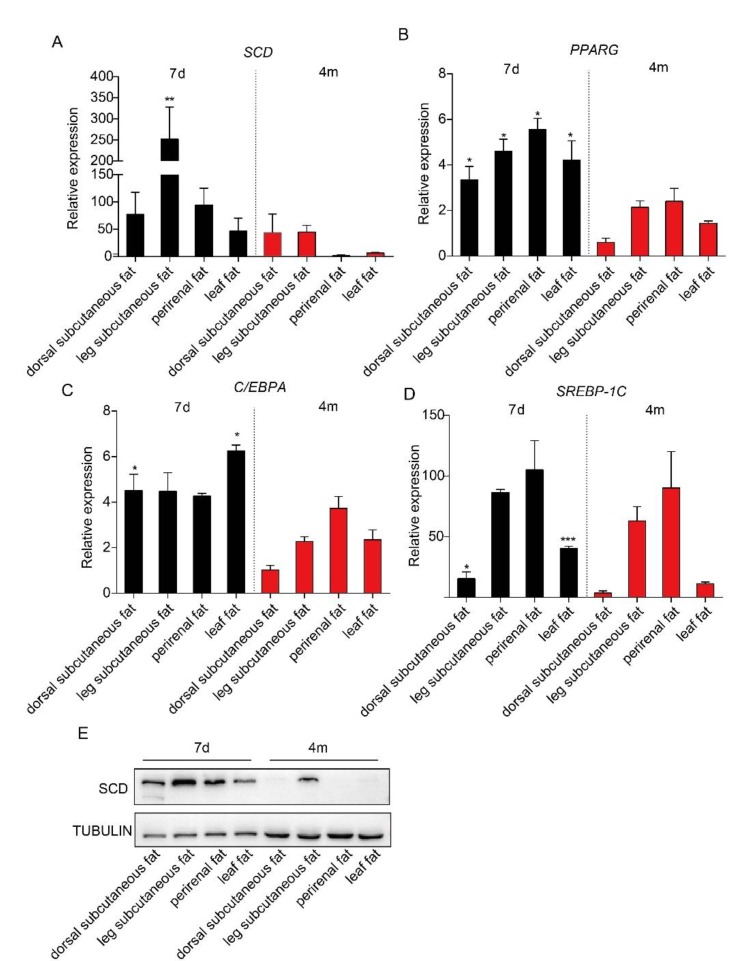
The expression level of stearoyl-CoA desaturase (SCD) and three transcriptional factors in various adipose depots from piglets and adult pigs. Subcutaneous adipose tissue from the dorsal and leg, perirenal adipose tissue, and leaf adipose tissue were collected from seven-day-old and four-month-old Bama pigs (*n* = 3/group). Real-time PCR results showed the expression levels of *SCD* (**A**), *PPARG* (**B**), *C/EBPA* (**C**) and *SREBP-1C* (**D**) in these adipose tissues. * *p* < 0.05, ** *p* < 0.01 and *** *p* < 0.001 were considered significant. (**E**) Western blot analysis results of SCD expression in the adipose tissues from seven-day-old piglets and four-month-old adult pigs.

**Figure 2 ijms-21-02446-f002:**
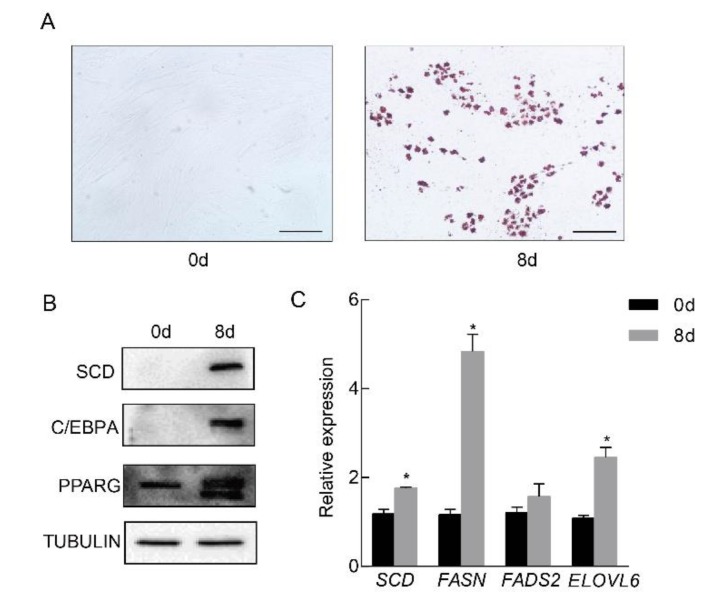
*SCD* was significantly up-regulated in differentiated porcine adipocytes. Porcine preadipocytes were primarily cultured and differentiated into mature adipocytes in vitro. (**A**) Left: nondifferentiated preadipocytes; right: oil red O staining of the differentiated cells. Positive staining indicated lipid droplet formation. Scale bar, 100 μm. (**B**) The expression levels of SCD and the adipocyte differentiation marker genes PPARG and C/EBPA in preadipocytes and differentiated adipocytes. TUBULIN was used as a loading control. (**C**) The expression levels of genes related to lipid synthesis were measured in preadipocytes and differentiated adipocytes (*n* = 3/group), * *p* < 0.05.

**Figure 3 ijms-21-02446-f003:**
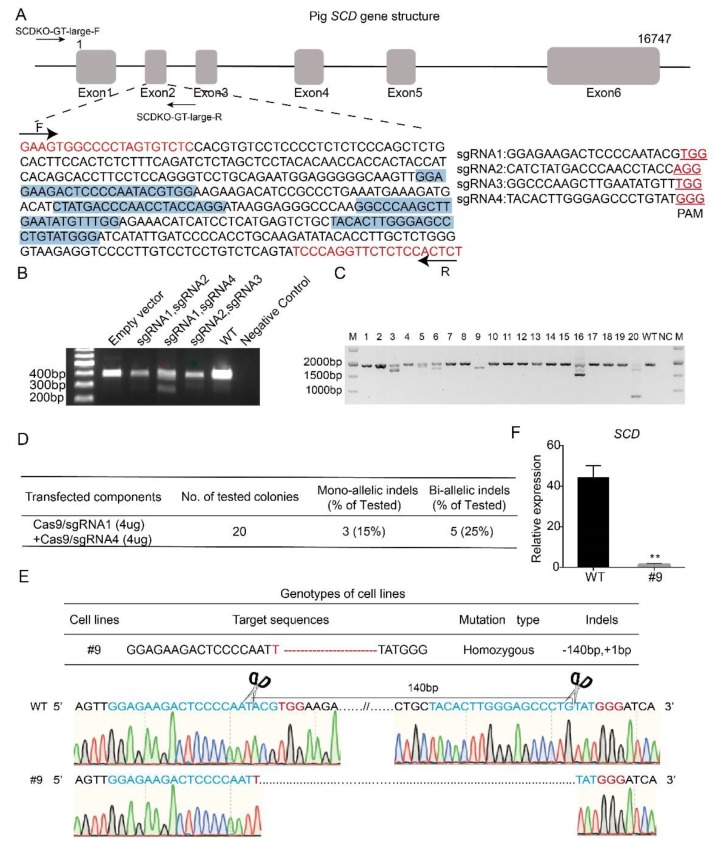
Generation of *SCD*^-/-^ porcine embryo fibroblasts. (**A**) Gene structure of the porcine *SCD* gene and the location of the four sgRNAs that target exon 2 (highlighted in blue). Primers designed to estimate the target efficiency are also indicated, and the predicted PCR size is 403 bp (highlighted in red). (**B**) Different pairs of vectors containing sgRNA, including sgRNA1-sgRNA2, sgRNA1-sgRNA4 and sgRNA2-sgRNA3, were selected for co-transfecting cells, because of a potential frame shift effect. A template with H_2_O was used as a negative control, and the DNA in the transfected empty vector was used as a positive control. Notably, the short band was observed in cells co-transfected with sgRNA1 and sgRNA4, and this pair of sgRNAs was then used for the remaining studies. (**C**) Plasmids containing sgRNA1 and sgRNA4 were co-transfected in porcine embryonic fibroblasts (PEFs), and single cells were sorted by flow cytometry and seeded into 96-well plates. A total of 20 single-cell colonies were selected for PCR and sequencing analysis. (**D**) Estimation of targeting efficiency. Notably, 5 biallelic mutant clones were identified, and the targeting efficiency was 25%. (**E**) Targeted sequence information of #9 colony. (**F**) Knockout efficiency of *SCD* in the #9 colony was examined by real-time PCR; ** *p* < 0.05.

**Figure 4 ijms-21-02446-f004:**
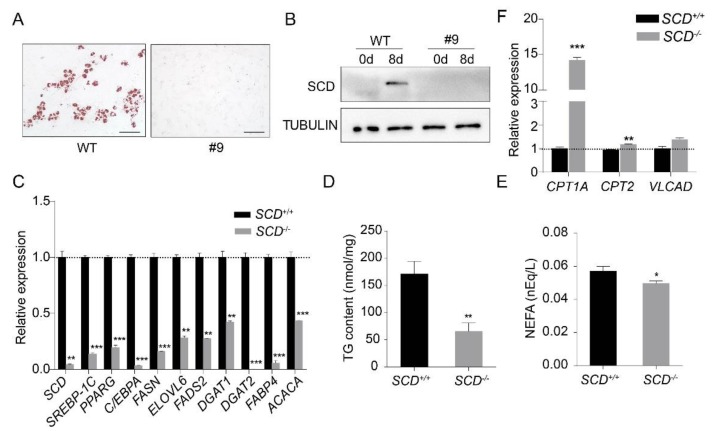
*SCD* knockout inhibited PEF transdifferentiation into mature adipocytes. (**A**) *SCD*^+/+^ and *SCD*^-/-^ were differentiated into adipocytes in vitro. Cells were subjected to oil red O staining after differentiation. Scale bar, 100 μm. (**B**) Western blot analysis of *SCD* in nondifferentiated and differentiated *SCD*^+/+^ and *SCD*^-/-^ cells. TUBULIN was used as the loading control. Notably, SCD was undetectable in differentiated *SCD*^-/-^ cells. (**C**) The expression levels of the genes related to lipid synthesis, in both the *SCD*^+/+^ and *SCD*^-/-^ cells. Notably, all of these genes were significantly lower in the *SCD*^-/-^ cells (*n* = 3/group). (**D**,**E**) TG (*n* = 7/group) and NEFA (*n* = 6/group) content levels in the *SCD*^+/+^ and *SCD*^-/-^ cells. (F) The expression levels of the genes related to fatty acid *β*-oxidation in both the *SCD*^+/+^ and *SCD*^-/-^ cells (*n* = 3/group); * *p* < 0.05, ** *p* < 0.01 and *** *p* < 0.001.

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
