# Peer review of "Stearoyl-CoA Desaturase Is Essential for Porcine Adipocyte Differentiation"

_ijms, 2020, doi:10.3390/ijms21072446_

Round 1
Reviewer 1 Report
IJMC 2020
Stearoyl-CoA desaturase is essential for porcine adipocyte differentiation
General comments
This manuscript describes SCD expression in porcine adipocytes and adipose tissue from pigs. The authors also describe the lack of differentiation of SCD knockout PEF, which is perhaps their most convincing set of experiments. I feel that the authors should have measured fatty acid composition of the adipose tissues, which would have confirmed functional SCD activity.
Specific comments
Abstract. Please state your hypothesis here and in the Introduction. What results did you expect to observe?
- 22 and throughout. Please change “fat” to “adipose tissue” except for its use in “body fat”. Fat is lipid containing an abundance of saturated fatty acids. Adipose tissue is what is being studied here.
- 33-36. The stated goal – the production of SCD-knockout pigs – would not be successful. Rodents have more than one SCD gene, so knocking out SCD1 was not lethal. Pigs (and cattle) express only one functional SCD gene, and knocking out that one SCD gene would disrupt all cell metabolism. I suggest reading Ntambi’s 2013 book, Stearoyl-CoA Desaturase Genes in Lipid Metabolism (The Springer Shop).
Also, the production of intramuscular adipose tissue in muscle is critical for meat quality. Even if the pigs survived, the end product (edible pork) would have inferior quality.
- 40-41. [1] is the wrong citation. There are decades of published research that describe the importance of carcass fat and muscle lipid for the pork production.
- 47-48. I believe this statement is incorrect. MUFA have no role in dietary unsaturated fat deficiency. You are referring to n-3 and n-6 polyunsaturated fatty acids.
- 48 and 51. Citations [6] and [7] are the same citation.
- 61 and 64. Citations [13] and [14] are the same citation.
- 82 and L. 126-127 (Figure 1 legend). This is incorrect. Only in leg s.c. adipose tissue was SCD expression higher in 7-d-old pigs than in 4-mo-old pigs.
- 85-86. The authors should include more details about the results in Figure 1. For example, the Western blot did not agree with the RNA data for 4-mo-old pigs.
- 133-134. The authors state “To our surprise…”. Why was this a surprise? Perhaps if they had stated a hypothesis, this could be considered a surprise.
Were the Landrace and Meishan pigs fed the same diet at the same location. SCD expression is very sensitive to diet, so if the pigs were fed different diet (even slightly different) or at different locations, these data are meaningless. This is especially true since no data were provided for fatty acid composition of the adipose tissue.
- 136-138. No mention is made of the results for FADS2.
L 162. This is incorrect. FADS2 was not different between breed types.
- 186 (Figure 3 legend). This is incorrect. SCD was not induced at high levels in differentiated porcine adipocytes. Although statistically different, SCD expression was only marginally greater than in undifferentiated adipocytes.
- 309. [21] is the wrong citation. It does provide data about meat quality etc.
- 326-328. I am not certain of the meaning of this sentence. It does not follow from the discussion on L. 320-326.
- 328-330. I don’t understand how your data from Landrace and Meishan pigs is consistent with the data from the young pigs. Weren’t they the same age and, again, were they fed the same diets at the same location?
- 343-348. Again, why was fatty acid composition not measured, which would have confirmed your supposition.
- 364-365. What was the source of adipose tissue from the Landrace and Meishan pigs?
Author Response
Response to Reviewer 1 Comments
General comments
This manuscript describes SCD expression in porcine adipocytes and adipose tissue from pigs. The authors also describe the lack of differentiation of SCD knockout PEF, which is perhaps their most convincing set of experiments. I feel that the authors should have measured fatty acid composition of the adipose tissues, which would have confirmed functional SCD activity.
Response: we appreciate this valuable comment. We agree with reviewer that fatty acid composition measurement could directly confirm the functional SCD activity. The fact is that the gene edited cells were grown up from the single cell and grew slowly, we tried to harvest more cells for the fatty acid composition measurement but failed. Despite this, our current data still provided the robust evidence to address the essential role of SCD in porcine adipogenesis.
Specific comments
- Please state your hypothesis here and in the Introduction. What results did you expect to observe?
Response: thanks for this good point. Based on suggestions of reviewer, we add our hypothesis in Abstract (line: 25-27) and Introduction (line: 70-73).
- 22 and throughout. Please change “fat” to “adipose tissue” except for its use in “body fat”. Fat is lipid containing an abundance of saturated fatty acids. Adipose tissue is what is being studied here.
Response: Based on the suggestions, we changed “fat” to “adipose tissue” for our samples in the revised manuscript (line 84-85 and Figure1 legend).
- 33-36. The stated goal – the production of SCD-knockout pigs – would not be successful. Rodents have more than one SCD gene, so knocking out SCD1 was not lethal. Pigs (and cattle) express only one functional SCD gene, and knocking out that one SCD gene would disrupt all cell metabolism. I suggest reading Ntambi’s 2013 book, Stearoyl-CoA Desaturase Genes in Lipid Metabolism (The Springer Shop).
Also, the production of intramuscular adipose tissue in muscle is critical for meat quality. Even if the pigs survived, the end product (edible pork) would have inferior quality.
Response: we appreciate very much for this valuable comment. We read the recommended book and agree that knockout of SCD in pig might be lethal. We make the revision on all related statement. We only focus on the effect of SCD on adipocyte differentiation in this manuscript (Line 35-37; line 369-372; line 375-379).
- 40-41. [1] is the wrong citation. There are decades of published research that describe the importance of carcass fat and muscle lipid for the pork production.
Response: sorry for this mistake. We made the revision in revised manuscript (please see line 43).
- 47-48. I believe this statement is incorrect. MUFA have no role in dietary unsaturated fat deficiency. You are referring to n-3 and n-6 polyunsaturated fatty acids.
Response: agree with reviewer, we had the incorrect statement here. It should be “SCD is required for protection against dietary unsaturated fatty acids deficiency”. We made the revision in revised manuscript (please see line 50).
- 48 and 51. Citations [6] and [7] are the same citation.
Response: we made the revision of citation in revised manuscript.
- 61 and 64. Citations [13] and [14] are the same citation.
Response: we made the correction in revised manuscript.
- 82 and L. 126-127 (Figure 1 legend). This is incorrect. Only in leg s.c. adipose tissue was SCD expression higher in 7-d-old pigs than in 4-mo-old pigs.
Response: thanks for this good point. We made the revision about this statement (line 85-95 and Figure 1 legend).
- 85-86. The authors should include more details about the results in Figure 1. For example, the Western blot did not agree with the RNA data for 4-mo-old pigs.
Response: We add more details about the results in Figure 1 (line 85-95). Actually, we made the comparisons of SCD expression in the same depot between the pigs at different ages, thus, our real-time PCR data are consistent with Western blot data basically. We guess reviewer mean the expression in dorsal s.c. depot, right? The reason for extremely lower expression by WB might be due to the specific individual, note that a bit bigger error bar in real-time PCR data. But this doesn’t affect the conclusion we made for Figure1A.
- 133-134. The authors state “To our surprise. Why was this a surprise? Perhaps if they had stated a hypothesis, this could be considered a surprise.
Response: thanks for this good point. It has been reported that global SCD1 knockout (KO) mice were protected against diet-induced adiposity, and liver-specific SCD1-KO mice were protected from high carbohydrate-induced obesity and hepatic steatosis because of reduced de novo lipogenesis and decreased fatty acid synthesis. Our unpublished data showed that the expression level of SCD is significant lower in lean animals, including mice and pig. Therefore, we would expect that adipose tissues from lean-type pig, Landrace, should contain the lower SCD expression. That is the reason we say “to our surprise”.
However, we agree with reviewer that the animal location and diet are very critical for SCD expression. The adipose tissues we used in this study for Figure 2 were kept in the lab, we track the record that they were raised at the same location, same diet. However, we have to say that these samples are not specifically for this study, and we know these pigs were not slaughtered at the same time. At this point, we are not sure whether the different batch of feed had the effect on SCD expression. We decided to delete this part in our manuscript.
Were the Landrace and Meishan pigs fed the same diet at the same location. SCD expression is very sensitive to diet, so if the pigs were fed different diet (even slightly different) or at different locations, these data are meaningless. This is especially true since no data were provided for fatty acid composition of the adipose tissue.
Response: As explained above, we deleted this piece of data in revised manuscript.
- 136-138. No mention is made of the results for FADS2.
Response: thanks for this good point and we made the revision of the result in revised manuscript (line 176-177).
L 162. This is incorrect. FADS2 was not different between breed types.
Response: We deleted this piece of data.
- 186 (Figure 3 legend). This is incorrect. SCD was not induced at high levels in differentiated porcine adipocytes. Although statistically different, SCD expression was only marginally greater than in undifferentiated adipocytes.
Response: thanks for this this comment. We revised the related parts to make our descriptions more accurate (line 198).
- 309. [21] is the wrong citation. It does provide data about meat quality etc.
Response: sorry that this mistake. We made the revision of the citations in revised manuscript.
- 326-328. I am not certain of the meaning of this sentence. It does not follow from the discussion on L. 320-326.
Response: thanks for this good point. We made the corresponding changes and some of sentences were deleted.
- 328-330. I don’t understand how your data from Landrace and Meishan pigs is consistent with the data from the young pigs. Weren’t they the same age and, again, were they fed the same diets at the same location?
Response: As explained above, we deleted this piece of data.
- 343-348. Again, why was fatty acid composition not measured, which would have confirmed your supposition.
Response: we agree with reviewer that fatty acid composition measurement will be greatly helpful to confirm our supposition in discussion. Actually, the gene edited cells were grown up from the single cell, we tried to harvest more cells for the fatty acid composition measurement but failed. Despite this, our data still provided the robust evidence to address the essential role of SCD in porcine adipogenesis. We will perform FA composition in the future. Thanks for this great comment again.
- 364-365. What was the source of adipose tissue from the Landrace and Meishan pigs?
Response: As explained above, we deleted this piece of data.
Reviewer 2 Report
This paper reports the role of SCD on adipocyte differentiation. SCD role in adipocyte differention has been extensively studied in mice and pigs, ref 22-25. This study describes other lipid genes involved and the difference in types of fat and pigs.
- Figure 1, line 85-86: It is not correct to state that 'these genes are highly expressed in adipose tissues from piglets‘. Q-RT-PCR results could only compare genes among adipose tissues from piglets at 7 d and 4 months. Best way is o compare these genes in adipose tissues and other tissues, e.g, liver.
- What is the reason that SCD gene is strongly suppressed in adipose of 4 months pigs compared with that of 7 d ? Authors should explaqin or speculate why in line 319.
Is it possible of age-dependence 7 d as pre-adipocytes and 4 months as mature-adipocytes. Not only SCD1 but also SCD2 play a role in differntiaiton of mouse 3T3-L1 preadipocytes (K H Kaestner, J M Ntambi, T J Kelly, Jr and M D Lane, JBC, 1989, 264, 14755-14761.)
- Figure 5F, how does SCD deficiency increse b-oxidaiton genes ? that results in decreased free fatty acids.
- Fig.3C, Fig.5C-F, number of samples= ? Fig. 5F legend (F) not (E).
- Line 344, mistake with (6)
- Line 346, please specifiy not just ..etc.
- Line 342, mispelling SREBF1.
- Line 462: grant number is missing.
Author Response
Response to Reviewer 2 Comments
Comments and Suggestions for Authors
This paper reports the role of SCD on adipocyte differentiation. SCD role in adipocyte differention has been extensively studied in mice and pigs, ref 22-25. This study describes other lipid genes involved and the difference in types of fat and pigs.
- Figure 1, line 85-86: It is not correct to state that 'these genes are highly expressed in adipose tissues from piglets‘. Q-RT-PCR results could only compare genes among adipose tissues from piglets at 7 d and 4 months. Best way is o compare these genes in adipose tissues and other tissues, e.g, liver.
Response: thanks for good point. We made the revision of the incorrect statement in revised manuscript and made the description in more detail (please find in line 85-95). In this study, we are interested in comparing the expression levels of SCD in different adipose depots, rather than in various tissues. Actually, it has been reported that adipose tissue had the significant higher expression of SCD, relative to liver1.
- What is the reason that SCD gene is strongly suppressed in adipose of 4 months pigs compared with that of 7 d ? Authors should explaqin or speculate why in line 319.
Response: Based on this suggestion, we added the possible reasons in “discussion part” (line 329-334).
- Is it possible of age-dependence 7 d as pre-adipocytes and 4 months as mature-adipocytes. Not only SCD1 but also SCD2 play a role in differntiaiton of mouse 3T3-L1 preadipocytes (K H Kaestner, J M Ntambi, T J Kelly, Jr and M D Lane, JBC, 1989, 264, 14755-14761.)
Response: Quite similar with comment above, based on our data from Figure1B and 1C, the expression levels of two adipoctye differentiation markers, PPARG and CEBPA, are higher in adipose depots from 7d piglets, suggesting the differentiation in those adipose depots has been initiated and are quite active, therefore consideration of adipose tissues from 7 d piglets as pre-adipocytes is not correct, because both genes don’t express in pre-adipocytes. However, adipose tissues from 4-month-old pigs can be thought as mature adipocytes, as most adipocytes are at the status of terminal differentiation.
We agree with reviewer, that both SCD1 and SCD2 were observed to be higher expressed in differentiation of mouse 3T3-L1 preadipocytes. Mouse Scd gene has four isoforms, Scd1, Scd2, Scd3 and Scd4. Scd1 and Scd2 expression is observed in adipose tissue. However, porcine SCD gene has two isoforms, SCD1 and SCD5. It has been clear that SCD1 was highly expressed in adipose tissues, while SCD5 mainly expressed in brain. Therefore, we only focus on SCD1 in our study.
- Figure 5F, how does SCD deficiency increse b-oxidaiton genes? that results in decreased free fatty acids.
Response: thanks for this good point. We found CPT1A was significantly and CPT2 was marginally upregulated in SCD knockout cells. It has been reported that SCD deficiency increases the levels of C18:0- or C16:0-CoAs, which could inhibit acetyl-CoA carboxylase (ACC). Low level of ACC results in the decrease of malonyl-CoA, therefore increases CPT12. We thought this might be the mechanism that b-oxidaiton genes were induced upon SCD knockout. However, we need to confirm experimentally. We tried to harvest more cells for fatty acid composition measurement, but failed due to few gene edited cells obtained from single cell screening. We will perform these experiments in the future to explore the detailed mechanisms regarding this question.
- Fig.3C, Fig.5C-F, number of samples= ? Fig. 5F legend (F) not (E).
Response: thanks for this good point. Fig.3C, n = 3/group. Fig.5C, n = 3/group. Fig.5D, n = 7/group. Fig.5E, n = 6/group. Fig.5F, n = 3/group. We added this information in Figure legend in the revised manuscript (line 205, 313-315).
- Line 344, mistake with (6)
Response: sorry for the mistake. We made the correction in revised manuscript. Please see the line 361.
- Line 346, please specifiy not just ..etc.
Response: based on the suggestion, we listed all of genes in the revised manuscript. Please see the line 363.
- Line 342, mispelling SREBF1.
Response: we made the correction for it (line 359).
- Line 462: grant number is missing.
Response: we put grant information in “Acknowledgements” part in old version. Based on the comment, we changed the grant information to “Funding” in revised manuscript. Please see the line 485-489.
References
- Ntambi; Ph., D.; James, M., Stearoyl-CoA Desaturase Genes in Lipid Metabolism. 2013.
- Ntambi, J. M.; Miyazaki, M.; Stoehr, J. P.; Lan, H.; Kendziorski, C. M.; Yandell, B. S.; Song, Y.; Cohen, P.; Friedman, J. M.; Attie, A. D., Loss of stearoyl-CoA desaturase-1 function protects mice against adiposity. Proc Natl Acad Sci U S A 2002, 99 (17), 11482-6.
Round 2
Reviewer 1 Report
Ijms-735094. Stearoyl-CoA Desaturase is Essential for Porcine 2 Adipocyte Differentiation
I appreciate the authors’ detailed response to my comments. They have revised their manuscript extensively and greatly improved its quality.
Minor comments
L. 50-51. On my original review, I indicated that “MUFA have no role in dietary unsaturated fat deficiency”. The authors changed “MUFA” to “SCD”, but this still is incorrect. The likely role of SCD (and MUFA) is to maintain membrane fluidity – the melting point of saturated fatty acids, palmitic and stearic acid, are too high (60 – 70°C) for normal cell function. This would explain why all cells express SCD to some extent, even those cells that do not accumulate lipid. MUFA cannot protect against dietary unsaturated fatty acid deficiency, as they cannot be converted to functional prostaglandins and leukotrienes.
L. 377-379. I located a fairly recent article that has the opposite goal – generation of transgenic animals to document the effects of the overexpression of SCD1 on obesity and steatosis. (citation listed below). The authors transfected porcine SK6 cells with a porcine SCD1 construct and increased the proportions of palmitoleic and oleic acid in the cells. Unfortunately, they did not report total lipid or expression of genes associated lipid deposition, as was done by the authors of this manuscript.
Lipids. 2018 Oct;53(10):933-945. doi: 10.1002/lipd.12102.
Author Response
Minor comments
- 50-51. On my original review, I indicated that “MUFA have no role in dietary unsaturated fat deficiency”. The authors changed “MUFA” to “SCD”, but this still is incorrect. The likely role of SCD (and MUFA) is to maintain membrane fluidity – the melting point of saturated fatty acids, palmitic and stearic acid, are too high (60 – 70°C) for normal cell function. This would explain why all cells express SCD to some extent, even those cells that do not accumulate lipid. MUFA cannot protect against dietary unsaturated fatty acid deficiency, as they cannot be converted to functional prostaglandins and leukotrienes.
Response: Thanks very much for explaining this point in detail. We agree that our original statement “SCD is required for protection against the dietary unsaturated fat deficiency” is not that accurate. We made the revision based on the reference that we cited here (Flowers, M. T.; Ntambi, J. M., Role of stearoyl-coenzyme A desaturase in regulating lipid metabolism. Current Opinion in Lipidology 2008, 19 (3), 248-256. Similar statement was in page 5). Please see line 52-55.
- 377-379. I located a fairly recent article that has the opposite goal – generation of transgenic animals to document the effects of the overexpression of SCD1 on obesity and steatosis. (citation listed below). The authors transfected porcine SK6 cells with a porcine SCD1 construct and increased the proportions of palmitoleic and oleic acid in the cells. Unfortunately, they did not report total lipid or expression of genes associated lipid deposition, as was done by the authors of this manuscript.
Response: Thanks for discussing the reference about overexpression of pig SCD1. Actually, we are happy to know that our data is kind of functionally consistent with data from listed paper. We knocked out SCD1 in PEF and try to use it as a candidate gene to decrease fat deposition in pigs, while they overexpressed SCD1 in porcine SK6 and aim to study its role in obesity and steatosis. As we described, the precise regulation of SCD activity needs to be explored for its potential application in both agriculture application or biomedical researches in the future. As similar statement was already in “Discussion” part, L.377-379 in “Conclusion” part was deleted in revise version. We appreciate very much for all reviewers’ comments and suggestions on our manuscript.